# Factors that Influence Sleep among Residents in Long-Term Care Facilities

**DOI:** 10.3390/ijerph17061889

**Published:** 2020-03-14

**Authors:** Da Eun Kim, Ju Young Yoon

**Affiliations:** 1School of Nursing, University of Wisconsin-Madison, Madison, WI 53705, USA; dkim622@wisc.edu; 2Research Institute of Nursing Science and College of Nursing, Seoul National University, Seoul 03080, Korea

**Keywords:** sleep, long-term care, aged, environment

## Abstract

Long-term care residents often experience sleep disturbances as they are vulnerable to a variety of physical, psychosocial, and environmental factors that contribute to sleep disturbances. However, few studies have examined the combined impact of multiple factors on sleep among long-term care residents. This study aimed to identify the factors that influence sleep efficiency and sleep quality based on a modified senescent sleep model. A total of 125 residents were recruited from seven long-term care facilities in South Korea. Sleep patterns and sleep quality were collected using 3-day sleep logs and the Minimal Insomnia Screening Scale for Korean adults (KMISS), respectively. The mean sleep efficiency was 84.6% and the mean score on sleep quality was 15.25. A multiple linear regression analysis showed that greater dependence in activities of daily living (ADL), higher pain, and light at night were related to lower sleep efficiency. Higher pain and fatigue, less activity time, noise and light at night, and lower nighttime staffing levels were related to poorer sleep quality. This study highlights that psychosocial and environmental factors as well as physical factors could influence sleep for long-term care residents. Our findings could be foundational evidence for multi-faceted sleep intervention program development in long-term care settings.

## 1. Introduction

Sleep is a physiological process with a recovery function that is essential for maintaining health [1]. The elderly living in long-term care facilities show more prevalent sleep disturbances and segmented sleep–wake patterns compared to community-dwelling older adults [2]. On average, long-term care residents are awake more than two hours at night [3], and more than one-third of residents have symptoms of insomnia [4]. In terms of sleep quality, about 72 percent of the elderly living in long-term care facilities report poor sleep quality [5]. Older adults in long-term care facilities are more likely to have chronic conditions that contribute to sleep disturbances and to have poor sleep hygiene than the general population of older adults, but the management of sleep disturbances in long-term care residents tends to be underappreciated [6]. Sleep disturbances in older adults can cause negative outcomes such as cardiovascular disease, cognitive decline, depression, and poor quality of life. In addition, daytime sleepiness due to sleep deprivation may increase cardiovascular disease, falls, and mortality [7]. Since sleep complaints at night can lead to daytime sleepiness and fatigue, the negative effects can continue throughout the day [1].

Sleep disturbances in the elderly are regarded as a multifactorial geriatric syndrome. One risk factor for sleep complaints is age-related changes, which are a part of the normal aging process. A decrease in the percentage of slow wave sleep and rapid eye movement (REM) sleep as well as a change in circadian rhythms appears with age [8]. Sleep disturbances are also more common in older adults with chronic or functional impairment such as urinary incontinence or pain [6]. Further, psychosocial and environmental factors such as reduced daytime activity, loneliness, excessive daytime napping, and inadequate sleep hygiene perpetuate sleep disturbances [6,7].

Long-term care residents are more vulnerable to a variety of risk factors for sleep complaints than community-dwelling older adults. Most residents have physical and/or cognitive functional impairment, which has a negative impact on sleep. They are also known to be mainly sedentary. According to an observation study, the time spent communicating with other people or engaging in leisure activities such as reading or playing games accounted for only 6% to 16% of the time observed [9]. Furthermore, since long-term care residents are institutionalized, they are likely to be exposed to noise or bright light at night, which can interfere with initiating or maintaining sleep [10]. According to one study comparing sleep patterns between older adults living in care facilities and those living in the community, the institutional environment is negatively associated with fragmented rest/wake patterns after controlling for the effects of participants’ individual characteristics [11]. Considering that sleep is associated with multiple factors including physical, psychosocial, and environmental, the factors influencing sleep among long-term care residents should be identified in depth. However, since previous studies have largely focused on the impact of a single factor on sleep, research on the combined impact of multiple factors on sleep is still lacking. There has been growing awareness of the importance of sleep management for the elderly living in long-term care facilities [12], but few studies have been conducted to comprehensively explore the factors affecting sleep patterns or sleep quality. One study reported that economic status and pain are negatively associated with sleep patterns among residents who live in South Korean long-term care facilities, but only demographic and physical characteristics were considered as explanatory variables [13]. In particular, little research has attempted to identify the impact of psychosocial and environmental factors on sleep patterns or sleep quality. The purpose of this study is to identify physical, psychosocial, and environmental factors influencing sleep patterns and sleep quality among older adults living in South Korean long-term care facilities based on a modified senescent sleep model.

## 2. Materials and Methods 

### 2.1. Study Design

This study used a cross-sectional design to identify the influencing factors related to sleep patterns and sleep quality among long-term care residents in South Korea (Korea hereafter) based on a modified senescent sleep model.

### 2.2. Study Participants

Participants were recruited using convenience sampling from seven long-term care facilities in Korea. Inclusion criteria for participation in the study were (1) aged 65 and older, (2) residence in a long-term care facility for more than one month, (3) capable of responding to the questionnaire and having a Korean Mini-Mental State Examination (K-MMSE) score of at least 15, and (4) voluntary agreement to participate in the research. According to the sample size calculator G-power program, version 3.1.9 (Dusseldorf University, Dusseldorf, Germany), 118 participants are needed for a sufficient sample for multiple linear regression analysis (effect size of 0.15, α error probability of 0.05, and power of 0.80) with 10 explanatory variables [14]. A total of 127 residents participated in this study, but two were excluded because of incomplete responses. Thus, 125 participants were included in the analysis.

### 2.3. Conceptual Framework

The senescent sleep model explains the mechanisms underlying sleep complaints among community-dwelling older adults [7]. According to the model, sleep disturbances can be caused by predisposing, precipitating, and perpetuating factors. Predisposing factors include changes in sleep–awake physiology due to the normal aging process. Precipitating factors include physical impairments that increase with aging. Perpetuating factors include psychosocial problems that persist sleep complaints. The conceptual framework of this study was developed to study influencing factors on sleep patterns and sleep quality in older adults living in long-term care facilities by modifying the senescent sleep model based on the literature review (Figure 1). In this conceptual framework, physical factors corresponding to the precipitating factors of the senescent sleep model consist of the activities of daily living (ADL), incontinence, pain, and fatigue [5,15]. Psychosocial factors, corresponding to the perpetuating factors of the senescent sleep model include loneliness, which is a perceived subjective feeling of being isolated [16], and activity time [7]. Environmental factors are also included in this conceptual framework since older adults who reside in a shared group residence facility can be more likely to be exposed to external environmental characteristics which can disrupt sleep. Noise and light at night, number of residents in a room, and staffing levels at night were studied as possible environmental factors [6,10,17].

### 2.4. Measures

#### 2.4.1. Sleep Patterns

Sleep patterns were assessed based on 3-day sleep logs. Sleep logs included the following entries for each day: time participant went to bed, sleep-onset latency, number of awakenings, duration of each awakening, and time participant woke up. Measurements obtained from the average of the 3-day sleep logs included the following [18,19]:Total bedtime (TBT): time from bedtime to getting out of bed;Total sleep time (TST): time in bed minus sleep-onset latency (SOL) and minus wake after sleep onset (WASO);Sleep-onset latency (SOL): time to fall asleep following bedtime;Number of awakenings (NWAK);Wake after sleep onset (WASO): the sum of wake times from sleep onset to the final awakening;Sleep efficiency (SE): TST divided by TBT times 100.

#### 2.4.2. Sleep Quality

Quality of sleep was measured using the Minimal Insomnia Screening Scale for Korean adults (KMISS) [20]. The KMISS is a self-report questionnaire consisting of the following three items: difficulty falling asleep at night, restlessness during sleep, and tiredness when waking up in the morning. The KMISS uses a summated rating scale with each item being scored on a scale of 0–10. The total scores ranged from 0 to 30 with higher scores indicating higher quality of sleep.

#### 2.4.3. Physical Factors

Activities of daily living (ADL) were measured by the Korean Activities of Daily Living (K-ADL) scale based on Katz’s Index [21]. The K-ADL includes seven items which measures physical function to perform basic tasks: dressing, washing face, bathing, feeding, transfer, toileting, and continence. Each item was measured on a 3-point Likert scale from 1 (independent) to 3 (dependent). A higher score indicated greater dependence in daily activities. Internal consistency reliability for this measure showed Cronbach’s α = 0.93 for the instrument development study [21] and Cronbach’s α = 0.83 for this sample. The weighted score developed by Won et al. [22] was calculated and used for regression analysis. In the calculation method of the weighted score, the item for continence was excluded as it was intended to investigate care needs rather than functional status [22]. The score ranged from 0 to 45. Incontinence was measured using one item for continence in the K-ADL with a 3-point Likert scale from 1 (independent) to 3 (dependent). Pain and fatigue were measured on an 11-point numerical rating scale (NRS). Higher scores indicated higher pain and fatigue intensity.

#### 2.4.4. Psychosocial Factors

Loneliness was measured using the 4-item Korean version of the Revised UCLA Loneliness Scale (RULS-4) [23,24]. Specifically, the RULS-4 consists of items 1, 13, 15, and 18 from among the total 20 items of the RULS. Each item is scored on a 4-point Likert scale from 1 (never) to 4 (often). Higher scores indicate higher loneliness. Internal consistency reliability was Cronbach’s α = 0.75 for the sample in which the measure was developed [24] and Cronbach’s α = 0.82 in this sample. Regarding activity time, we assessed the amount of the time which engaged in individual/group informal activity time and formal activity time based on the activity theory [25]. The individual/group informal activity time includes the time for solitary activities such as reading or watching TV, and the time for informal activities with staff, other residents, and family. The formal activity time includes participation in group activity programs in the facilities. In the regression analysis, total activity time (i.e., the sum of individual/group informal activity time and formal activity time) was used as a representative indicator of activity time.

#### 2.4.5. Environmental Factors

Nighttime noise was measured using a noise recognition scale for long-term care facilities. This measurement was a modified version of the noise recognition scale for intensive care unit settings [26] based on the classification of the sources of noise in long-term care facilities [17]. The noise recognition scale for long-term care facilities is a self-report questionnaire measuring the extent to which sleep was disturbed by noise. It included six items (i.e., staff vocalization, resident vocalization, television or radio, telephones or alarms, equipment, and bathroom) and was scored using a 4-point Likert scale from 1 (not at all disturbed) to 4 (very disturbed). Light at night was measured by one item, asking how much nighttime sleep was disturbed by light. The item was scored on a 4-point Likert scale from 1 (not at all disturbed) to 4 (very disturbed). For each source of noise and light, a score of 3 or 4 indicated that noise and light interfered with nighttime sleep. A score of 1 or 2 indicated that noise and light did not interfere with nighttime sleep. The number of residents in a room is the number of residents sharing a room including the participant and all other residents who share the room with the participant. Staffing levels at night were determined by the ratio of total number of residents to the number of direct care workers working during the nighttime.

#### 2.4.6. Demographic Characteristics 

The demographics of the participants included gender, age, educational attainment, marital status, religion, cognitive function, long-term care grade, length of stay, taking sleep medications or sedatives, and participating in sleep intervention programs. Cognitive function was assessed based on the Korean Mini-Mental State Examination (K-MMSE) [27]. The K-MMSE consists of 30 items using a binary (0 or 1) scale. The total score ranges from 0 to 30, with higher scores indicating higher cognitive function. If the participant had taken the K-MMSE in the last six months, that score was used to minimize participant fatigue from duplicate test participation.

### 2.5. Data Collection

Data were collected from July to September in 2017 in seven long-term care facilities that allowed data collection. The researchers explained the purpose and procedures of the study to the chief directors of the participating long-term care facilities. Once the chief directors approved, the nursing managers in the facilities gave the research team a list of residents who were potentially eligible to participate in the study. The researcher then approached each resident independently and explained the purpose and procedures of this study in detail. Those who voluntarily agreed to participate in the research signed a written informed consent. The survey was completed by interviewing participants in person and collecting participants’ responses to the survey. After completing the survey, participants were instructed on how to complete their sleep logs. Direct care workers and nursing staff were also trained on how to complete the sleep logs. If necessary, direct care workers helped the participants complete their sleep logs using interviews right before going to bed and immediately after waking up.

### 2.6. Data Analysis 

We used descriptive statistics to analyze participant demographics, as well as sleep-related, physical, psychosocial, and environmental characteristics. Multiple linear regression analyses were performed to identify factors influencing sleep patterns and sleep quality. We conducted the Shapiro–Wilk test to investigate the normality of the dependent variables, including both sleep efficiency and sleep quality. Neither of these results was unsatisfactory. Subsequently, robust maximum-likelihood (RML) estimation was used to statistically adjust for nonnormality. Because convenience sampling was used, there was a risk that statistical independence may not have been maintained among the participants living in the same facility. The intracluster correlation (ICC) coefficient of sleep efficiency was ρ = 0.120 and that of sleep quality was ρ = 0.097 [28]. Multi-level analysis was used for multiple linear regressions to statistically adjust for intracluster correlation. The multiple linear regression analyses were conducted using Mplus 7.0 (Muthén & Muthén, Los Angeles, CA, USA), and the remaining analyses were conducted with IBM SPSS Statistics 24.0 (IBM Corp., Armonk, NY, USA).

### 2.7. Ethical Considerations

This study was conducted under the ethical approval of the Seoul National University Institutional Review Board (IRB No. 1706/002–013). The purpose and processes of this study were explained to potential participants in detail. Participants were informed about and assured of the anonymity and confidentiality of their responses. Participants were also informed that they could decline to participate or withdraw from participation at any time. Written consent was then obtained only from those who voluntarily agreed to participate in this study prior to data collection. 

## 3. Results

### 3.1. General Characteristics of the Study Participants

The majority of participants were female (72.8%), with a mean age of 82.1 years (Table 1). Approximately 76.0% of participants were widowed and the most common education level was “finished elementary school” (67.2%). The majority of participants did not report a religion (61.6%), and the mean cognitive function scores measured by K-MMSE were 20.62. The mean length of stay in the present facility was 40.37 months. The care grade of the sample included 46 (36.8%) 3rd grade level long-term care residents and 50 (40.0%) 4th grade level residents. Eighteen participants (14.4%) took sleep medications or sedatives (e.g., anti-anxiety agents, antidepressants, antipsychotics). None of them had previously participated in sleep intervention programs.

### 3.2. Physical, Psychosocial, and Environmental Characteristics of the Study Participants

#### 3.2.1. Physical Characteristics

The mean weighted ADL score measured by the K-ADL was 17.28 (SD = 9.00; Table 2). Regarding incontinence, 12 (9.6%) were totally dependent, 29 (23.2%) were partially dependent, and 84 (67.2%) were independent. The mean score for pain intensity was 3.47 (SD = 3.12) and that for fatigue intensity was 3.86 (SD = 2.96).

#### 3.2.2. Psychosocial Characteristics

The mean score for loneliness measured by the RULS-4 was 10.78 (SD = 2.64; Table 2). The number of participants who engaged in individual or group informal activities was 115 (92.0%) and the mean daily informal activity time was 143.76 min (SD = 74.01). In addition, 94 (75.2%) participated in formal activities, and the mean formal activity time in a day was 25.92 min (SD = 17.60).

#### 3.2.3. Environmental Characteristics

Resident vocalization (46.4%), staff vocalization (34.4%), and bathroom (10.4%) noise were the major sources of nighttime noise. The number of participants responding that nighttime sleep was disturbed by any type of noise was 66 (52.8%). Forty-seven (37.6%) participants responded that nighttime sleep was disturbed by light at night. The mean number of residents who lived in the same room (including the participant) was 3.06 (SD = 1.04). For all seven long-term care facilities, no nursing staff worked at night, but the average number of residents per direct care worker at night was 10.24 (SD = 3.11).

### 3.3. Sleep-Related Characteristics of the Study Participants

The mean TBT was 502.60 min (SD = 59.09) and the mean TST was 424.17 min (SD67.18; Table 3). The mean SOL was 28.61 min (SD = 13.68), and the average number of awakenings during sleep was 2.20 times (SD = 0.81). The mean WASO was 47.34 min (SD = 21.93) and the mean sleep efficiency was 84.6% (SD = 6.38). The mean time to go to bed was 8:55 p.m. and the mean time to wake up was 5:18 a.m. The mean score on sleep quality measured by the KMISS was 15.25 (SD = 6.21).

### 3.4. Factors Influencing Sleep Patterns and Sleep Quality

A multi-linear regression analysis was performed on two models, one with sleep patterns as a dependent variable and the other with sleep quality as a dependent variable. The model for sleep patterns used sleep efficiency, a measure that reflects the total sleep time, sleep-onset latency, and awakening time during nighttime sleep, as a representative indicator. Based on the conceptual framework of this study, (1) physical factors included ADL, incontinence, pain, and fatigue; (2) psychosocial factors included loneliness and activity time; and (3) environmental factors included nighttime noise and light, the number of residents in the same room, and staffing levels at night. The variance inflation factor (VIF) values for all independent variables ranged from 1.18 to 2.11, which indicated that there was no multicollinearity present in the regression model. The Durbin–Watson statistic was used to test the autocorrelation of the error term. The value for the sleep efficiency model was 1.84 and that for the sleep quality model was 1.71, which indicated no autocorrelation of the error term in both models.

#### 3.4.1. Sleep Efficiency

In terms of the physical factors, participants with poorer ADL function (β = -0.19, *p* = 0.042) and those with higher pain (β = −0.26, *p* = 0.001) reported lower sleep efficiency (Table 4). None of the psychosocial factors was significantly associated with sleep efficiency. Among the environmental factors, participants who responded sleep was disturbed by light at night (β = −0.23, *p* = 0.007) showed lower sleep efficiency. The R-square value was 0.176 (*p* < 0.001), which means 17.6% of the variation in sleep efficiency can be explained by the independent variables.

#### 3.4.2. Sleep Quality

Among the physical factors, participants with higher pain (β = −0.22, *p* < 0.001) and those with higher fatigue (β = −0.15, p = 0.004) were more likely to report poorer sleep quality (Table 4). In terms of the psychosocial factors, participants who engaged in activities for less time (β = 0.16, *p* = 0.007) reported poorer sleep quality. Among the environmental factors, noise at night (β = −0.20, p < 0.001), light at night (β = −0.16, *p* < 0.001), and lower staffing levels at night (β = −0.18, *p* = 0.049) were significantly associated with poorer sleep quality. The R-square value was 0.398 (*p* < 0.001), which means 39.8% of the variation in sleep quality can be explained by the independent variables.

## 4. Discussion

We examined physical, psychosocial, and environmental factors that influence sleep patterns and sleep quality based on a modified senescent sleep model considering the characteristics of the elderly living in long-term care facilities [7]. The senescent sleep model proposed that physical factors such as loss of physical function or poor health status resulting from usual aging precipitate sleep disturbances [7]. In this study, the results showed that dependent ADL function and higher pain are associated with lower sleep efficiency among residents living in long-term care facilities. Likewise, participants with higher pain and those with higher fatigue reported poorer sleep quality. Long-term care residents who are not able to perform activities of daily living independently have limitations engaging independently in activities that contribute to improved sleep [10]. Further, long-term care residents with physical dependence spend a longer time in their beds, which negatively affects sleep [3]. Pain can also cause arousal of the cerebral cortex, making it difficult to initiate sleep and maintain sleep [29]. A study using electroencephalograms reported that muscle and joint pain decreases the delta waves occurring during slow wave sleep and increases the alpha and beta waves occurring during shallow sleep or activity [30]. In terms of fatigue, the results of this study are similar to the results reported in other studies of older adults [18] or patients who have had strokes [31]. In particular, patients with chronic fatigue syndrome are known to have an increased risk of non-restorative sleep and restlessness [32] and 81% of patients with chronic fatigue syndrome reported at least one sleep disorder [33]. Therefore, care should be taken to enable residents to maintain ADL function along with active pain and fatigue management to promote sleep for the elderly living in long-term care facilities.

Regarding psychosocial factors, we found more engagement in activities was associated with higher sleep quality among residents living in long-term care facilities. This result is similar to the result of a previous study which showed that engaging in activities such as mild physical activity and playing games has positive effects on subjective sleep quality among older adults in the U.S. [34]. Activity is related to an internal biological sleep-wake mechanism, the circadian rhythm [35]. Structured schedules of activity during the daytime can serve as external time cues (Zeitgeber) to continuously regulate the sleep–wake rhythm [35]. In addition, engagement in activities provides a sense of belonging and companionship. It is known that these supportive social relationships enable older adults’ psychological well-being, which eventually can improve nighttime sleep [36]. However, an observation study reported long-term care residents slept or did nothing in 36% to 62% of the observations from 7 a.m. to 11 p.m. [9]. Therefore, engagement in activities should be encouraged to improve residents’ sleep quality.

In this study, we added environmental factors from the senescent sleep model [7], taking into account that the study participants are institutionalized. The results indicated that light exposure at night was associated with lower sleep efficiency among long-term care residents. In addition, nighttime noise and light and lower staffing levels at night were associated with poorer sleep quality. Light exposure suppresses melatonin secretion, which prevents synchronizing circadian rhythms and increases the number of awakenings [37]. Regarding noise, noise-induced stress negatively affects sleep quality by interacting with other environmental and personal stresses [38]. Specifically, among the sources of noise, residents said that resident vocalization (46.4%) and staff vocalization (34.4%) interfered with nighttime sleep most frequently. These findings are particularly noteworthy because noise from staff and other residents and light exposure at night are modifiable by improving the institutional policy to minimize nighttime noise and light. Intervention studies have identified the effects of reducing nighttime noise, light, and sleep-disruptive nursing care practices on sleep in U.S. long-term care facilities [39,40]. For example, an intervention including using quiet shoes and small flashlights instead of overhead room lights and providing incontinence care when residents are awake showed increased sleep efficiency [40]. In Korea, however, few studies have been developed to minimize nighttime noise and light in long-term care settings. Based on the results of this study, further research is needed to develop intervention programs for nighttime sleep hygiene management in long-term care settings. In terms of staffing levels at night, a low staffing level leads to heavy workloads and low quality of care, contributing to dependent residents spending a longer time in bed [3]. Likewise, bedtimes and getting-up times for dependent residents were influenced by staff shift patterns [3]. The staffing standard for Korean long-term care facilities is 2.5 residents per direct care worker, but night shift staffing is not regulated. Thus, it is necessary to set up minimum standards for night shift staffing to improve care quality for residents’ sleep.

Several limitations should be addressed in this study. First, since this study is a cross-sectional study, it is difficult to identify causal relationships between sleep and explanatory variables. Second, residents who had cognitive impairment and difficulty communicating were excluded. Thus, the results of this study cannot be generalized to all long-term care residents. Last, except for the use of sleep medications or sedatives, other medications that may influence sleep were not included in the study.

None of the long-term care facilities participating in this study had a sleep intervention program. Since multiple risk factors are associated with low sleep efficiency and poor sleep quality among long-term care residents, multi-component strategies combining two or more categories of treatment are recommended as an effective therapy, such as social and physical stimulation, clinical care practice, and environmental intervention [19,41]. Particularly in Korea, some researchers have conducted limited single sleep intervention programs, such as bright light therapy or aromatherapy, in long-term care settings [42,43]; however, little research has been conducted on the development of multi-component sleep intervention programs. A systematic review suggested that multidisciplinary interventions such as combining minimization of nighttime noise, reduction of nursing care at night, and participation in daytime activities are most effective for improving sleep [44]. Long-term care residents have different health needs; thus, individualized sleep approaches should be provided. However, most of the sleep-related care is provided mainly in a task-oriented manner [3]. Based on the findings of this study, we suggest that individualized multi-component sleep intervention programs should be developed according to physical, psychosocial, and environmental needs of individuals.

## 5. Conclusions

Our findings indicate that multiple psychosocial and environmental factors as well as physical factors have a significant impact on sleep among older adults living in long-term care facilities. While it is not easy to improve long-term care residents’ physical health due to aging and underlying disease, psychosocial and environmental factors are relatively modifiable by promoting engagement in activities and improving institutional policies to manage sleep hygiene. These results can contribute to developing individualized multi-component sleep interventions to improve sleep patterns and sleep quality most effectively. Future longitudinal studies are needed to examine the causal relationship between sleep and associated variables.

## Figures and Tables

**Figure 1 ijerph-17-01889-f001:**
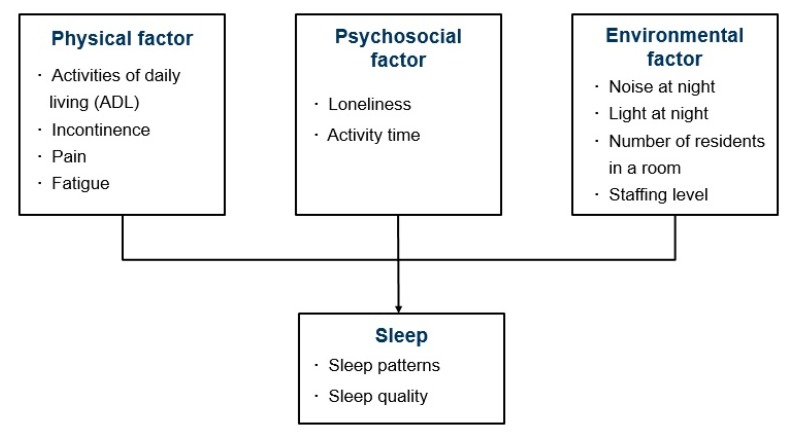
Conceptual framework of this study.

**Table 1 ijerph-17-01889-t001:** General characteristics of the study participants (*n* = 125).

Variables.	Categories	*n* (%)	Mean (SD)
Gender	Male	34 (27.2)	
	Female	91 (72.8)	
Age (years)			82.14 (7.37)
Level of education	None	21 (16.8)	
	Elementary school	84 (67.2)	
	Middle school	7 (5.6)	
	High school or more	13 (10.4)	
Marital status	Married	21 (16.8)	
	Widowed	95 (76.0)	
	Divorced	8 (6.4)	
	Unmarried	1 (0.8)	
Religion	Yes	48 (38.4)	
	No	77 (61.6)	
K-MMSE			20.62 (3.55)
Length of stay (months)			40.37 (40.37)
Long-term care grade	1	3 (2.4)	
	2	20 (16.0)	
	3	46 (36.8)	
	4	50 (40.0)	
	5	2 (1.6)	
	Not eligible	4 (3.2)	
Sleep medication or sedative use	Yes	18 (14.4)	
	No	107 (85.6)	
Sleep interventions	Yes	0 (0.0)	
	No	125 (100.0)	

K-MMSE: Korean Mini-Mental State Examination.

**Table 2 ijerph-17-01889-t002:** Physical, psychosocial, environmental characteristics of the participants (*n* = 125).

Figure	Variables	Categories	*n* (%)	Mean (SD)
Physical	ADL (K-ADL [weighted])			17.28 (9.00)
	Incontinence	Independent	84 (67.2)	
		Partially dependent	29 (23.2)	
		Totally dependent	12 (9.6)	
	Pain			3.47 (3.12)
	Fatigue			3.86 (2.96)
Psychosocial	Loneliness (RULS-4)			10.78 (2.64)
	Activity time (min)	Informal activity (individual/group)	115 (92.0)	143.76 (74.01)
		Formal activity	94 (75.2)	25.92 (17.60)
Environmental	Source of nighttime noise	Resident vocalization	58 (46.4)	
		Staff vocalization	43 (34.4)	
		Bathroom	13 (10.4)	
		Equipment	7 (5.6)	
		Bell or alarm	4 (3.2)	
		Television or radio	3 (2.4)	
	Any source of noise		66 (52.8)	
	Light at night		47 (37.6)	
	Number of residents in a room			3.06 (1.04)
	Staffing level at night			10.24 (3.11)

ADL: activities of daily living; K-ADL: Korean Activities of Daily Living scale; RULS-4: 4-item Revised UCLA Loneliness Scale.

**Table 3 ijerph-17-01889-t003:** Sleep-related characteristics of the study participants (*n* = 125).

Variables.	Categories	Mean (SD)
Sleep patterns	TBT (min)	502.60 (59.09)
	TST (min)	424.17 (67.18)
	SOL (min)	28.61 (13.68)
	NWAK	2.20 (0.81)
	WASO (min)	47.34 (21.93)
	SE (%)	84.61 (6.38)
	Time to go to bed (h:m)	PM 8:55
	Time to wake up (h:m)	AM 5:18
Sleep quality (KMISS)		15.25 (6.21)

TBT: total bedtime; TST: total sleep time; SOL: sleep-onset latency; NWAK: number of awakenings; WASO: wake after sleep onset; SE: sleep efficiency; KMISS: Minimal Insomnia Screening Scale for Korean adults.

**Table 4 ijerph-17-01889-t004:** Factors influencing sleep efficiency and sleep quality based on multiple linear regression analysis.

Factors	Variables	Sleep Efficiency (*n* = 120)			Sleep Quality (*n* = 125)		
		*β*	*SE*	*p*	*β*	*SE*	*p*
Physical factors	ADL	−0.19	0.10	0.042	−0.18	0.13	0.144
	Incontinence	−0.02	0.12	0.868	0.08	0.08	0.310
	Pain	−0.26	0.08	0.001	−0.22	0.05	<0.001
	Fatigue	−0.04	0.09	0.688	−0.15	0.05	0.004
Psychosocial factors	Loneliness	0.06	0.05	0.250	0.03	0.05	0.491
	Activity time	−0.05	0.06	0.414	0.16	0.06	0.007
Environmental factors	Noise at night	0.07	0.10	0.445	−0.20	0.04	<0.001
	Light at night	−0.23	0.08	0.007	−0.16	0.05	<0.001
	^#^ Resident	0.16	0.12	0.177	−0.01	0.15	0.970
	Staffing level at night	0.08	0.11	0.456	−0.18	0.09	0.049

ADL: activities of daily living; ^#^ Resident: number of residents in a room; β: standardized coefficient; SE: standard error.

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
