# Peer review of "Factors that Influence Sleep among Residents in Long-Term Care Facilities"

_ijerph, 2020, doi:10.3390/ijerph17061889_

Round 1

Reviewer 1 Report

Mental health of old people are worthy of interest. It is also interesting to see a scientific report performed in non western countries, allowing comparison of data. Unfortunately, some major points should be address to improve this manuscript. see the following:

1) Introduction.

It was not clear to me why the authors choose people living in long term care facilities. Their vulnerability to sleep disturbances is higher than old people in the general population? Is there a difference in socioeconomic status between old people inside and outside facilities? These points should be clarified.

2) Methods.

Here my main concern is the application of the senescent sleep model to this study. This model have a lot of gap and it is hardly acceptable the items inside the 3 boxes which may affects sleep. Many things should be discussed or justified:

*I don't understand why number of residents and staffing level are in environmental factors while they are usually used as proxy measure for neighborhood deprivation or social position

*loneliness is a consequence of living conditions in industrialized country, not necessary a psychological factor related to sleep. To my opinion, loneliness induces anxiety which in turn expose individual to sleep complains. Loneliness can also be an environmental factor because it can be influenced by the social network or proximity of family members. And, an individual may developed sleep disturbances even without loneliness...

*why pain and fatigue are used as physical factors when it is well documented that someone in long-term facility care will not necessary have them, or sleep complains appears without pain and fatigue.

*i don't see the practical difference between Activities of daily living and activity time

*where is the place of acute or chronic stress 

This model needs a lot of improvement, it is not a solid conceptual framework and should be used with caution.  I suggest to justify methodological choices for this paper.

3) Discussion.

the authors discussed a lot about stress, sleep-wake cycle and anxiety. The problem is, they were not introduced and justified in previous sections. This lake of justification create some paragraphs which are not linked in a logical flow. Also due to the reasons explained in 2), sentences like ''Psychosocial factors negatively influence the sleep–wake cycle and such influences may magnify the effect of physical factors and perpetuate sleep complaints'' are not supported by theorical background of this paper neither by its results.

I suggest to rewrite this section with more references in support to arguments stated here.

Author Response

Thank you for your great comments and suggestions. We answered all the points you made below. 

Point 1: Introduction: It was not clear to me why the authors choose people living in long term care facilities. Their vulnerability to sleep disturbances is higher than old people in the general population? Is there a difference in socioeconomic status between old people inside and outside facilities? These points should be clarified. 

→ Response 1: Thank you for your comment. As you suggested, we clearly added several sentences to describe long-term care residents’ vulnerabilities to sleep disturbances compared to general population of older adults (Line 32-35, 47-48, 54-57)

Point 2: Methods: *I don't understand why number of residents and staffing level are in environmental factors while they are usually used as proxy measure for neighborhood deprivation or social position

→ Response 2: Nursing homes are living places for residents, but they are quite different environments compared to the community homes in terms of consistent provision of care by staff under the mostly shared space. Due to unique environmental context, noise, lighting and room sharing were included in environmental factors which can interrupt sleep in long-term care setting in a previous study (Ye & Richards., 2018). We also assumed that shared-living environment could be an interrupting factor to sleep. Particularly, residents are more likely to feel noisy or other discomfort that makes it difficult to initiate or maintain sleep when they share bedroom with other residents. Regarding staffing level, low staffing level at night are associated with longer time spent in bed and a lack of resident choices about bedtime (Luff et al., 2011). With this regards, we included these variables in external environmental factors. We added the word ‘external’ to describe it more clearly (Line 102-104).

* Ye, L.; Richards, K. C. Sleep and long-term care. Sleep Med. Clin. 2018, 13, 117-125; DOI: 10.1016/j.jsmc.2017.09.011.

* Luff, R.; Ellmers, T.; Eyers, I.; Young, E.; Arber, S. Time spent in bed at night by care-home residents: choice or compromise? Ageing & Soc. 2011, 31, 1229-1250.

Point 3: Methods: loneliness is a consequence of living conditions in industrialized country, not necessary a psychological factor related to sleep. To my opinion, loneliness induces anxiety which in turn expose individual to sleep complains. Loneliness can also be an environmental factor because it can be influenced by the social network or proximity of family members. And, an individual may developed sleep disturbances even without loneliness...

Response 3: We totally agree with your opinion that loneliness can be associated with social network or family proximity. However, according to a recent consensus study report, social isolation and loneliness are two distinct phenomena (National Academies of Sciences, 2020). Social isolation is a structural indicator, but loneliness is a perceived subjective feeling. Although those who are socially isolated may feel lonely, social isolation and loneliness often are not significantly correlated (National Academies of Sciences, 2020). Also, in the senescent sleep model (Vaz fragoso & Gill, 2009), loneliness is already included in psychosocial factors which can perpetuate sleep complaints. For these reasons, we would like to keep loneliness in the psychosocial factors.

* National Academies of Sciences, Engineering, Medicine. Social isolation and loneliness in older adults. 2020. Available from: http://www.nationalacademies.org/hmd/Reports/2020/social-isolation-and-loneliness-in-older-adults

* Vaz Fragoso, C. A.; Gill, T. M. Sleep complaints in community‐living older persons: A multifactorial geriatric syndrome. J. Am. Geriatr. Soc. 2007, 55, 1853-1866; DOI: 10.1016/j.archger.2012.11.011.

Point 4: Methods: why pain and fatigue are used as physical factors when it is well documented that someone in long-term facility care will not necessary have them, or sleep complains appears without pain and fatigue.

→ Response 4: Previous studies reported that pain and fatigue are significantly associated with sleep complaints among older adults. Accordingly, we hypothesized that participants with higher pain and those with higher fatigue would show lower sleep efficiency and poorer sleep quality. In this study, we found that higher pain was significantly associated with lower sleep efficiency and poorer sleep quality. Likewise, higher fatigue was significantly associated with poorer sleep quality.

Point 5: Methods: i don't see the practical difference between Activities of daily living and activity time

Response 5: Thank you for your comment. Activities of daily living (ADL) and the word of activity in the activity time is different. ADL is a variable which primarily measures physical functional ability to perform basic self-care tasks: dressing, washing face, bathing, feeding, transfer, and toileting. Activity time refers to the amount of time engaged in leisure activities such as individual/group informal activity time (e.g., watching TV, listening to music, or talking with other residents) and formal activity time based on the activity theory. We added some explanation to describe the difference between the two variables clearly (Line 133, 151).

Point 6: Methods: where is the place of acute or chronic stress; This model needs a lot of improvement, it is not a solid conceptual framework and should be used with caution.  I suggest to justify methodological choices for this paper.

Response 6: The conceptual framework in this study was developed based on the senescent sleep model, a geriatric version of an insomnia model (Vaz fragoso & Gill, 2009). Some previous studies among young adults or workers reported associations between sleep and stress, but in case of older adults living in long-term care facilities, there are few studies that consider stress as a sleep-related factor. Given that the participant of this study is long-term care residents, we did not include stress in our framework. However, as your suggestion, we will consider to included stress as a perpetuating factor for the future study.

* Vaz Fragoso, C. A.; Gill, T. M. Sleep complaints in community‐living older persons: A multifactorial geriatric syndrome. J. Am. Geriatr. Soc. 2007, 55, 1853-1866; DOI: 10.1016/j.archger.2012.11.011.

Point 7: Discussion: the authors discussed a lot about stress, sleep-wake cycle and anxiety. The problem is, they were not introduced and justified in previous sections. This lake of justification create some paragraphs which are not linked in a logical flow. Also due to the reasons explained in 2), sentences like ''Psychosocial factors negatively influence the sleep–wake cycle and such influences may magnify the effect of physical factors and perpetuate sleep complaints'' are not supported by theorical background of this paper neither by its results. I suggest to rewrite this section with more references in support to arguments stated here.

→ Response 7: Thank you for your great suggestion. When we described mechanisms linking sleep-related factors to low sleep efficiency and poor sleep quality, stress, sleep-wake cycle and anxiety were mentioned. As you suggested, we deleted several sentences which are not linked in a logical flow and revised this section focusing on the logical relationships based on the conceptual framework (Line 308-329).

Reviewer 2 Report

The manuscript is about identifying the underlying factors responsible for the sleep problems among the residents in the long-term care facilities in South Korea. The authors collected the sleep data by using sleep logs for 3 days from elderly residents (>65 years old). The criteria used to identify sleep problems were sleep patterns (efficiency) and sleep quality based on a modified senescent sleep model. The authors found that these residents had a mean sleep efficiency of 84.6% and the mean score on sleep quality was 15.25. The independent variables chosen were Physical, psychosocial and environmental. This is a very well written manuscript. However, I have one minor concern: The sleep efficiency of 84.6% in people of ages >65 is not uncommon and is normal at this age. My question is that if this is due to age or other precipitating factors.

Author Response

Point 1: I have one minor concern: The sleep efficiency of 84.6% in people of ages >65 is not uncommon and is normal at this age. My question is that if this is due to age or other precipitating factors.

→ Response 1: Thank you for your comment. The sleep efficiency among residents in long-term care facilities in this study (84.6%) was similar to that among community-dwelling older adults in previous studies. The sleep efficiency of 84.6% was slightly lower than that of 85%, which usually indicates one of the criteria of insomnia. We believe that normal aging process as well as precipitating factors could have negative influences on sleep disturbances in this setting. With this regards, this study was conducted based on the modified senescent sleep model to identify multiple factors that influence sleep efficiency and sleep quality among residents living in long-term care facilities.

Round 2

Reviewer 1 Report

.